# The Role of Autophagy and Autophagy Receptor NDP52 in Microbial Infections

**DOI:** 10.3390/ijms21062008

**Published:** 2020-03-16

**Authors:** Shuangqi Fan, Keke Wu, Mengpo Zhao, Erpeng Zhu, Shengming Ma, Yuming Chen, Hongxing Ding, Lin Yi, Mingqiu Zhao, Jinding Chen

**Affiliations:** 1College of Veterinary Medicine, South China Agricultural University, No. 483, Wushan Road, Tianhe District, Guangzhou 510642, China; hnndfsq@126.com (S.F.); 13660662837@163.com (K.W.); huananweishengwu@163.com (M.Z.); zhu13782701756@126.com (E.Z.); mashengming@stu.scau.edu.cn (S.M.); cym27@sina.cn (Y.C.); dinghx@scau.edu.cn (H.D.); yilin@scau.edu.cn (L.Y.); 2Guangdong Laboratory for Lingnan Modern Agriculture, Guangzhou 510642, China

**Keywords:** autophagy, autophagy receptor, ubiquitin-proteasome, NDP52, microbial infection

## Abstract

Autophagy is a general protective mechanism for maintaining homeostasis in eukaryotic cells, regulating cellular metabolism, and promoting cell survival by degrading and recycling cellular components under stress conditions. The degradation pathway that is mediated by autophagy receptors is called selective autophagy, also named as xenophagy. Autophagy receptor NDP52 acts as a ‘bridge’ between autophagy and the ubiquitin-proteasome system, and it also plays an important role in the process of selective autophagy. Pathogenic microbial infections cause various diseases in both humans and animals, posing a great threat to public health. Increasing evidence has revealed that autophagy and autophagy receptors are involved in the life cycle of pathogenic microbial infections. The interaction between autophagy receptor and pathogenic microorganism not only affects the replication of these microorganisms in the host cell, but it also affects the host’s immune system. This review aims to discuss the effects of autophagy on pathogenic microbial infection and replication, and summarizes the mechanisms by which autophagy receptors interact with microorganisms. While considering the role of autophagy receptors in microbial infection, NDP52 might be a potential target for developing effective therapies to treat pathogenic microbial infections.

## 1. Introduction

Macroautophagy (hereafter referred to as autophagy) is an internal balancing mechanism for maintaining homeostasis in eukaryotic cells. After receiving an autophagy induction signal, such as pathogen infection [1], starvation [2], growth factor withdrawal [3], endoplasmic reticulum (ER) stress, and oxidative stress [4,5], a small double-membraned structure forms in the cytosol and then expands to form a bowl structure (commonly termed phagophore) consisting of lipid bilayers that can be observed under electron microscopy. Many components in the cytoplasm, including protein aggregates, damaged organelles, and foreign invading pathogens, are enclosed within the forming “vesicle”, which becomes a closed spherical autophagosome [6]. LC3, which is also known as microtubule-associated protein 1 light chain 3 (Map1lc3), consists of two interconvertible forms (LC3-I and LC3-II) and is involved in the formation of autophagosome membranes. Subsequently, the autophagosome fuses with lysosomes to form autophagolysosome, where the "autophagic substrate" in autophagosomes are degraded by various hydrolases (Figure 1) [7,8].

Autophagy selectively removes damaged or aging organelles to maintain proteostasis and intracellular homeostasis, playing a significant role in the occurrence, development, and outcome of infectious diseases through various mechanisms of pathogen clearance. Accumulating evidence has revealed that autophagy acts as a cellular immune defense strategy for inhibiting the growth of pathogenic microorganisms or removing pathogenic microorganisms from the host cell [9]. Therefore, it is not surprising that pathogenic microorganisms have evolutionarily developed specific or nonspecific mechanisms to evade or employ autophagy for effective replication and pathogenesis [10]. For example, *Helicobacter pylori*, through secretions (Vacuolating cytotoxin A, VacA) and lipopolysaccharide (LPS) on the bacterium, can activate and promote autophagy. The autophagic vesicles that formed are adapted for the multiplication of *H. pylori* in the host [11]. Some microorganisms have evolved different strategies to escape from (*M. tuberculosis*) [12], inhibit (HIV) [13] or use autophagy (HBV) to sustain their infections in the hosts [14]. Our previous studies have revealed that classical swine fever virus (CSFV) also exploits autophagy to promote its replication [15,16]. Besides, the relationship between certain pathogenic microorganisms and autophagy is more complicated. In innate immunity, autophagy contributes to the pattern response downstream of the effector receptor, which in turn promotes phagocytosis [17]. Autophagy and Toll-like receptors (TLRs), NOD-like receptors (NLRs), and RIG-I-like receptors (RLRs) have mutual regulation [18,19]. For example, different TLRs (including TLR-1, TLR-2, TLR-3, TLR-5, and TLR-6) are activated by their ligands to promote autophagy by interacting with Beclin1 via Myd88 or Trif [20], thus contributing to the clearance of pathogenic microorganisms and enhancing the protective effect of the host [21]. The NLRs recognize peptidoglycans in bacterial cell walls, and Nod1 or Nod2 can activate autophagy to resist bacterial invasion through interaction with ATG16L1 [22,23]. The ATG5-ATG12 complex can directly interact with the CARD domain of RIG-I and IPS1 to suppress the downstream RLRs signal [24], which suggests that ATG proteins may act as negative regulators of RLR-mediated antiviral response.

NDP52 is a member of the nuclear dots (NDs) protein family [25], which contains a SKICH30 domain at the amino terminus [26], the central region is a coiled-coil domain, and the carboxy terminus consists of one LIM domain and two zinc finger motifs [27]. The zinc finger motifs contain the binding domain for ubiquitin (Figure 2). NDP52, which is also known as Calcoco2 (calcium binding and coiled-coil domain 2), plays important roles in ubiquitin-mediated protein degradation. It is well-known that the ubiquitin-proteasome system (UPS) is another major degradation system in host cells. After ubiquitination, invasive pathogens may be cleared by the host cells via autophagy [28]. Autophagy receptors, such as NDP52 and SQSTM1, play intermediate roles in autophagy and ubiquitination [29]. Ubiquitin acts as a signaling molecule and plays an important role in xenophagy. On one hand, the autophagy receptor proteins recognize and bind to autophagic substrates in a ubiquitin binding domain-dependent or independent manner. On the other hand, the LC3-interaction region (LIR) in autophagy receptors allows for their binding to the ATG8 family members (such as LC3) that are located at autophagosomes [30]. After the fusion of autophagosome with lysosome, the substrates are degraded by acid hydrolases within the lysosome [31]. Selective autophagy might be differentially regulated, as ubiquitin chains have different forms of linkage. Studies have shown that the K63-linked polyubiquitin chain can mediate the degradation of substrate proteins via autophagy [32,33]. Once entrapped within an autophagosome, bacteria may survive or escape, otherwise they will be rapidly destroyed [34]. Autophagosome maturation allows for the discharge of lysosomal enzymes in autolysosomes, leading to the destruction of the bacteria. Similar to the Traf6-binding protein (T6BP), the NDP52 protein is also a binding partner of myosin VI, and it is involved in the regulation of cytokine signaling and cell adhesion by binding to myosin VI [35,36]. NDP52 controls autophagosome maturation by interacting with ATG8 orthologs and myosin VI, thus mediating autophagic degradation [37].

Autophagy acts as a “double-edged sword”, which generally functions as a pro-survival strategy for host cells; nevertheless, it also can be hijacked by other microorganisms to promote their infection and pathogenesis in host cells [38]. The relationship between multiple pathogenic microorganisms and autophagy is complex. The role of NDP52 in pathogenic microorganisms has attracted increasing attention from researchers [39]. Apart from its significant role in autophagic degradation, NDP52 is also involved in the host defense mechanisms [40]. In this review, we focus on the relationship between autophagy receptor NDP52 and microbial infection, and provide new insight into the mechanism of pathogenic microbial infection and development of potential anti-infection methods.

## 2. The Role and Mechanism of Autophagy and NDP52 in Bacterial Infections

Autophagy plays an important role in bacterial infections. First, as a defense system for host cells, autophagy is activated during bacterial invasion, and the invading bacteria are removed by encapsulating and delivering them to lysosomes. However, bacteria have also developed several mechanisms to deal with the clearance by autophagy during the long-term evolution. The autophagy receptor NDP52 plays an important role in the autophagy-bacteria interplay. Researchers performed a structure-based prediction of protein interactions and discovered that the proteins of *Salmonella*, *Listeria* and *Shigella* can modify the selective autophagy receptors SQSTM1 and NDP52. These findings indicate a close correlation between autophagy receptor NDP52 and bacterial infection [41]. Next, we will focus on the relationship between autophagy and bacteria, such as *Salmonella*, *Listeria*, *Shigella*, and *Streptococcus*, and reveal the role of autophagy in bacterial infection and clearance, as well as the specific mechanism of NDP52 in bacterial infection (Table 1). 

## 3. The Role of NDP52 in *Salmonella* Infections

*Salmonella* is a genus of Gram-negative enterobacteria, which is the causative agent of salmonellosis. Salmonella enteritidis serotype *Salmonella typhi* (*S. typhi*) and *Salmonella paratyphi* can cause typhoid fever in humans, especially in Asia and Africa, where there is a lack of clean water and proper sanitation [45]. Infection with non-typhoidal *Salmonella* (NTS) serovars, such as *S. enterica serovar Typhimurium*, and *S. enteriditis* also cause a significant disease burden, with an estimated 93.8 million cases worldwide and 155,000 deaths each year, posing a great threat to human health [46]. Moreover, the treatment of Salmonella infection has become challenging, owing to the emergence of drug-resistant strains [47]. 

*Salmonella typhimurium* serotypes invade host cells and colonize ”*Salmonella*-containing vesicles (SCV)” for self-replication after the vesicle rupture, the bacteria are released into the cytoplasm and then multiply [48]. However, there is little SCV ruptured, mainly due to the expression of TANK-binding kinase 1 (Tbk1) in the host cell. Tbk1 maintains the integrity of the SCV follicle membrane, preventing SCV follicle swelling and rupture, thereby reducing the number of bacteria entering the cytosol [49]. At the same time, the host cells also generate a series of innate immune responses to clear the pathogens that enter the cytoplasm, of which the most important response is selective autophagy. Besides, autophagy recognizes intracellular *S. enterica serovar Typhimurium* in damaged vacuoles [50]. Selective autophagy protects the cytoplasm from invading bacteria, during which NDP52 plays a center role in the recognition and binding to bacteria [51]. Fip200 and SINTBAD/NAP1 are subunits of the Ulk and Tbk1kinase complex, respectively, which are independently recruited to *Salmonella* by NDP52 [52]. When the trimeric complex is combined, “eat-me” signal initiates autophagy, thereby enclosing the bacteria into the autophagic vesicles [53]. When *Salmonella* is released into the cytoplasm from the SCV, the bacteria are modified by poly-ubiquitination and NDP52 binds the bacterial ubiquitin coat as well as ATG8/LC3, and delivers cytosolic bacteria into autophagosomes. NDP52 not only plays an important role in the recognition of the invaded *Salmonella*, but also in the maturation of autophagosomes encapsulating *Salmonella*. 

NDP52 interacts with LC3a, LC3b, and/or Gabarapl2 through a distinct LC3-interacting region, interacting with myosin VI to promote the maturation of autophagosomes. NDP52 for autophagosome maturation is complementary, but it does not depend on its function in pathogen-targeted autophagosomes, which depends on interaction with LC3c (Figure 3) [37].

## 4. The Role of NDP52 in *Shigella* Infections

*Shigella* is a type of Gram-negative bacillus as the most common pathogen causing human bacterial dysentery in developing countries [54]. During *Shigella* entry, Nod1/2 initiates autophagy by detecting bacterial peptidoglycan and recruiting ATG16L [55]. In LC3-related phagocytosis (LAP), a subset of autophagy proteins (e.g., LC3) are recruited to the phagosome membrane and promote fusion with lysosomes. NADPH oxidase activity plays an important role in LAP and the activation of the ATG conjugation systems, which is dispensable for xenophagy [56]. Interestingly, during *Listeria* and *Salmonella* infection, LAP and xenophagy both occur at the same time [57], and they can be difficult to distinguish, since both are defined by membrane-associated LC3 [52]. In the cytosol, actin-polymerized bacteria are recognized by ATG5 and embedded in the septin cage structure, which prompts the bacteria to target autophagy degradation, and prevents its spreading [58]. *Shigella flexneri* has developed a mechanism to escape autophagy in the cytoplasm. In the cytosol, *Shigella* prevents ATG5 from recognizing IcsA and septin in cage, by expressing IcsB [59]. The researchers showed that the autophagy protein ATG5 initiates autophagy by binding to the IcsA protein on the surface of bacteria, however, in the presence of another *Shigella* protein IcsB, it competitively binds to the ATG5 site with IcsA, to protect bacteria from autophagy degradation [60].

The survival replication of dysentery bacilli in host cells depends on the key effector protein VirA, which is secreted by the type III system. In the early stages of *Shigella* infection (40 min), the type III secretes effector protein IcsB and recruits Toca-1 to intracellular bacteria, which prevents the recruitment of LC3 and other autophagy machinery. The IcsB also inhibits the Toca-1 interaction with LC3, and the suppression of NDP52 occurs synchronously directly [61]. Another mechanism by which *Shigella flexneri* inhibits autophagosome formation in the cytosol is the secretion of protein VirA to inactivate Rab1. *Shigella* dysenteriae and enteropathogenic *Escherichia coli* utilize a class of virulence effector molecules to mimic the mode of action of host TBC-like GAP to inactivate the host small G protein Rab1 specifically, and finally inhibit the anti-infective defense pathway that is mediated by host autophagy pathway, and inflammatory factor secretion. Some cytosolic *Shigella* disrupt the host actin cytoskeleton formation and advance the actin tail to spread between cells to escape the cytosolic immune response [62]. NDP52-mediated selective autophagy also plays an important role in the process of *Shigella* infection [29,42]. The researchers found that *Shigella*-septin cages recruit NDP52, which colocalizes with *Shigella* in its vicinity. Further, studies have found that ubiquitin-binding adaptor proteins SQSTM1 and NDP52 targeted *Shigella* in an autophagy pathway that is dependent upon septin and actin. Interestingly, in *Shigella* infection, IcsB not only prevents autophagy, but also prevents ubiquitin-protein recruitment/formation and the recruitment of SQSTM1 and NDP52 [43].

NDP52 also plays an important role in the related diseases that are caused by *Shigella*. Crohn’s disease is an inflammatory bowel disease that is characterized by changes in the intestinal microbiome [63]. There is an increased likelihood of infection by foodborne pathogens, such as *Salmonella* and *Shigella* in the intestine of patients with Crohn’s disease and a large number of studies have shown an association between mutations in NDP52 and Crohn’s disease [64]. During *Salmonella* infection, anti-bacteria autophagy requires the participation of Tbk1 and WIPI-2 [65], and NDP52 initiates selective autophagy through the recruitment of Ulkand Tbk1 kinase complexes [66]. In addition, NDP52 can promote the maturation of Salmonella-encapsulated vesicles to promote *Salmonella* clearance [34]. *Shigella* recruits Toca-1 to inhibit the aggregation of NDP52 and LC3 to reduce xenophagy [67]. In addition, *Shigella* alters the NF-κB signaling pathway and inflammatory response to promote its own replication [68]. Studies have found that NDP52 promotes NF-κB activation [57], which suggests that NDP52-regulated *Salmonella* and *Shigella* replication may be used as a therapeutic means of Crohn’s disease. 

## 5. The Role of NDP52 in *Listeria* Infections

*Listeria* is a Gram-positive facultatively intracellular foodborne pathogen that is often found in food and elsewhere in nature [69]. Serious *Listeria* poisoning might cause blood and brain tissue infections [70]. *Listeria* is also an intracellular living flora that is similar to *Shigella* [71]. *Listeria* monocytogenes rapidly replicates in the cytoplasm after infecting the host. The researchers found that the bacteria were colonized in the cytoplasm while using a cholesterol-dependent listeriolysin O (LLO) to escape phagocytosis [72]. While LLO activity is not sufficient to allow for bacteria to escape phagocytosis, the bacteria are surrounded by phagosomes and form spacious *Listeria*-containing phagosomes (SLAP), being generated in an autophagy dependent manner allowing for bacteria to grow slowly. SLAPs keep the bacteria and the host immune system in a paralyzed state; hence, the formation of SLAP is considered to be a mechanism of bacterial persistent infection [73]. *Listeria* infected mouse embryonic fibroblasts (MEF) cells can induce autophagy in the early stages of infection [74], and intracellular infection might significantly induce mitophagy in macrophages. Furthermore, studies have found that *Listeria* monocytogenes induces calcium influx, mitochondrial damage, and mitochondrial autophagy by the secretion of LLO. Autophagy plays an important role in the early stages of *Listeria* infection by limiting the growth of *Listeria* and inhibiting the formation of SLAP [75]. *Listeria* monocytogenes evades growth restrictions from autophagy by regulating PrfA and phospholipases [76]. ActA is required, but not sufficient, for the escape of *Listeria* monocytogenes from autophagy, in the cytoplasm [77]. Similarly, NDP52-mediated selective autophagy also plays an important role in *Listeria* infection. Previous studies have exploited *Shigella* and *Listeria* as intra-cytosolic tools for characterizing different pathways of selective autophagy. The authors showed that the ubiquitin-binding adaptor proteins SQSTM1 and NDP52 target *Shigella* to an autophagy pathway dependent upon septin and actin. In contrast, SQSTM1 or NDP52 targeted the *Listeria* ActA mutant in an autophagy pathway that was independent of septin or actin [43]. 

## 6. The Role of NDP52 in *Streptococcus* Infections

*Streptococcus* is a Gram-positive bacterium, but might lose its Gram-positive character in old or pus specimens. *Streptococci* does not usually cause disease, but *Group A Streptococcus* (GAS) is invasive strain that can cause significant mortality [78,79]. Autophagy plays an important role in streptococcal infection and clears out GAS infected in non-phagocytic cells. GAS that escape from endosomes to cytoplasm are wrapped in autophagosome-like compartments and are eliminated upon the fusion of these compartments with lysosomes. Moreover, in the ATG5-deficient cells, GAS survive, multiply, and release from the cells [80]. NDP52-mediated selective autophagy regulates bacterial growth. Researchers found that NDP52 binds the bacterial ubiquitin coat as well as ATG8/LC3 and delivers cytosolic bacteria into autophagosomes that affect the bacterial growth. Knockout the NDP52 and Tbk1 (IKK family kinase colocalizing with NDP52 at the bacterial surface) in cells, *Streptococci* replicated well and were aggregated by the ubiquitin-encapsulated bacteria ATG/LC3-positive autophagosomes [44]. Further research indicated that this effect of NDP52 is closely related to Rab35 GTPase. The researchers revealed that Rab35 GTPase recruits NDP52 to *Streptococcus*, thereby initiating selective autophagy and degrading *Streptococci*. The expression of autophagy receptors is inhibited in the early stage of wild-type *Streptococcal* infection. The inhibitor of Rab35 GTPase, TBC1D10A, inhibited the expression of autophagy and NDP52, but Rab35 GTPase could bind to NDP52 in a Tbk1-dependent manner, thereby recruiting NDP52 to *Streptococcus*, promoting the maturation of autophagic vesicles and the degradation of bacteria [81]. However, *Streptococcus* has also evolved a corresponding mechanism to escape the degradation by autophagy. The researchers found that the M1T1 clones of GAS could replicate well in cells and escape autophagy degradation, which was related to the expression of SpeB, a streptococcal cysteine protease. SpeB inhibited the expression of autophagy receptors, such as SQSTM1, NDP52, and Nbr1 in cells avoiding ubiquitin and the recognition of the host autophagy molecule LC3, thereby evading host autophagy and promoting its replication [82]. Moreover, NDP52 works in conjunction with LC3-associated phagosome (LAP) in Streptococcus pneumoniae infection. *Streptococcus pneumoniae* can trigger the formation of pneumococcus-containing LC3-associated phagosome (LAPosome)-like vacuoles (PcLVs) in an early stage of infection, which are indispensable for the subsequent formation of bactericidal pneumococcus-containing autophagic vacuoles (PcAVs). The ATG16L1 WD domain, SQSTM1, NDP52, and poly-Ub, contributed to PcLV formation [83].

## 7. The Role and Mechanism of NDP52 in Viral Infections

The original study revealed that the location of NDP52 was reset during virus infection and interferon therapy [84]. In herpes simplex virus-1 (HSV-1) infection, NDP52 was removed from the nucleus by the virus. In the early stages of adenovirus type 5 (Ad5) infection, NDP52 was isolated in 2–3 μm length tracks, and then removed from these tracks and accumulated in the outer rim adjacent to the virus replication area in the later stages of infection. The research also found that IFN-γ treatment improved the expression and localization of NDP52 in the cytoplasm [84]. These results indicate that NDP52 plays a key role in the viral cycle, and we summarized the role of autophagy and NDP52 in virus infection (Table 2). 

During the infection of the influenza virus, viral protein PB1-F2 plays an important role in influencing the viral virulence by inducing immune cell apoptosis and enhancing inflammation [85]. The researchers found that the influenza virus protein PB1-F2 interacts with NDP52 to regulate innate immunity by activating NF-κB and type I interferon-associated pathways in a Traf6-dependent manner. Further studies showed that the co-expression of NDP52, Mavs, and PB1-F2 enhanced the inflammatory response, which indicated that NDP52 is involved in the regulation of influenza virus replication through the PB1-F2 protein [86]. 

Measles virus (MeV) is an enveloped virus with a negative-stranded RNA genome [87], which can induce autophagy during viral infection, thereby promoting its replication [88,89]. Studies have shown that NDP52 plays an important role in regulating viral replication by binding to MeV-C and MeV-V proteins in MeV-infected cells [90]. Furthermore, the inhibition of NDP52 significantly reduces MeV replication not through the entry of the virus, but the maturation of autophagosome. Interestingly, although NDP52 can inhibit the NF-κB signaling pathway [91], its replication effect on MeV is independent of either the NF-κB signaling pathway or T6BP. In another study, the researchers found that the Z matrix proteins of Lassa virus (LASV) and Mopeia virus (MOPV) can interact with NDP52, but NDP52 does not affect MOPV and LASV replication [92].

In the Coxsackievirus B3 (CVB3) infection, the researchers found that during function, NDP52 is cleaved by the viral protein 3C, which results in a stable C-terminal fragment, but this fragment maintains the full-length gene function of NDP52. NDP52 and SQSTM1 differentially regulate the life cycle of the virus, inhibiting SQSTM1 to promote CVB3 replication, but inhibiting NDP52 to reduce CVB3 replication. Although NDP52 and SQSTM1 can both interact with the viral protein VP1 and promote the ubiquitination of the VP1 protein, only NDP52 can inhibit the production of type I interferon by promoting the Mavs protein [93]. In Chikungunya virus (CHIV) infection, human NDP52, but not mouse-derived NDP52, can interact with the viral protein nsp2, which can significantly reduce CHIV replication and inhibit NDP52 expression. Virus-induced cell death, which has an important role in viral replication, and NDP52 reduces virus-induced cell death, which indicates that NDP52 regulates viral replication by acting on cell death [94]. Similarly, NDP52 interacts with the Tetherin protein and plays an important role in mediating IFN. Tetherin promotes the viral replication of VSV and Sendai virus (SeV), and reduces IFN production. Further studies have found that this regulation is closely related to the ubiquitin of antiviral protein Mavs, and NDP52 can reduce Mavsand de-ubiquitination, as well as its interaction with Tetherin protein [95]. 

Our study found that NDP52 plays an important role in the infection of CSFV. CSFV belongs to the genus *Pestivirus* within the family *Flaviviridae*. The virus has a small, enveloped, single-stranded, positive-sense 12.3-kb RNA genome with a long, open reading frame that encodes a 3898-amino acid polypeptide [96]. Co- and post-translational processing of the polypeptide by cellular and viral proteases yields 12 cleavage products, including four structural proteins (C, Erns, E1, and E2) and eight non-structural proteins (Npro, p7, NS2, NS3, NS4A, NS4B, NS5A, and NS5B). CSFV can infect several cell types, including immune cells, which leads to cellular immunosuppression [97]. However, CSFV infection does not cause a cytopathic effect, and the underlying infection mechanisms remain unclear [98,99]. We found that CSFV infection inhibits autophagy receptor NDP52 expression, ubiquitination, and SUMO2-4 modification. Further studies revealed that CSFV mediates the ubiquitination and SUMOylation of NDP52 via the PINK1/Parkin pathway. Moreover, the inhibition of NDP52 decreases the induction of mitophagy and inhibits CSFV replication. The inhibition of NDP52 reduces CD63 expression and it is binding to the CSFV E2 protein, which has an essential role in persistent CSFV infection. We investigated whether NDP52 inhibits CSFV replication through the release of immune factors and antivirus signals since NDP52 has a close relationship with the NF-κB innate immunity pathway and an important role in the antiviral response. Our results show that inhibiting NDP52 boosts interferon and release and promotes NF-κB pathway activation. In summary, we found that NDP52 inhibition not only reduces CSFV binding and entry into autophagic vesicles, but also inhibits CSFV replication by active NF-κB antiviral immune pathways [100] (Figure 4). Our data reveal a novel mechanism by which NDP52, an autophagy receptor, mediates CSFV infection, and provides new avenues for the development of antiviral strategies.

In summary, autophagy plays an important role in the survival and replication of pathogenic microorganisms. With the deepening of research, the use of regulating autophagy as a treatment method has attracted increasing researchers’ attention. The autophagy receptor protein NDP52 has also received increasing attention, it not only serves as a link connecting autophagy substrates with lysosomes, but also acts as a ”bridge“ that closely links the two degradation pathways of ubiquitin-proteasome system and autophagy. However, NDP52 plays a different role in bacterial and viral infections. For example, NDP52-mediated autophagosome maturation promotes *Salmonella* clearance, but promotes MeV replication. Although the NDP52 function is mostly related to autophagy, its function is not limited to mediating the autophagic degradation of substrates. NDP52 is also closely related to the NF-κB signaling pathway and Mavs antiviral protein, but, so far, their interactions and their role in viral infection have been studied less. Therefore, the complete molecular mechanism of action of the autophagy receptors in autophagy and other signaling pathways needs to be further elucidated to lay a solid foundation for clarifying the mechanism of microbial infection. 

## Figures and Tables

**Figure 1 ijms-21-02008-f001:**
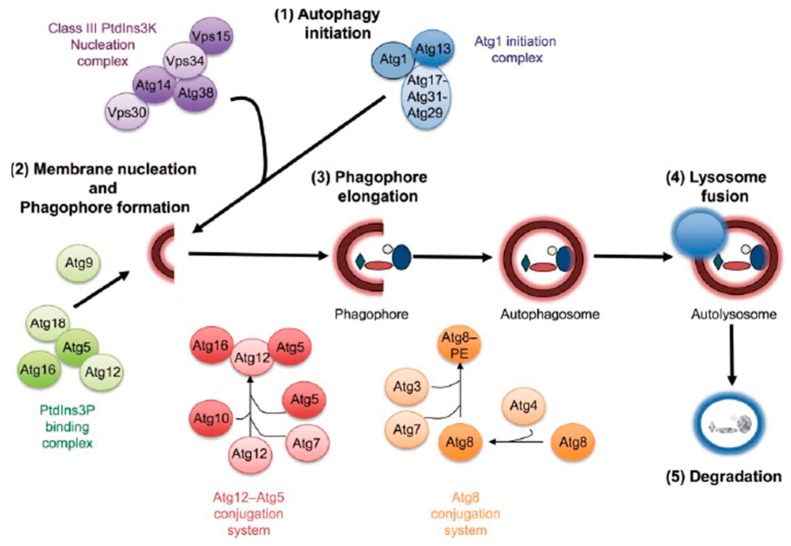
The autophagic process. Autophagy proceeds through at least five discrete steps: initiation, membrane nucleation and phagophore formation, phagophore elongation, lysosome fusion, and degradation. These steps are controlled by at least five different functional groups of proteins: ATG1/Ulk1 initiation complex, class III PtdIns 3-kinase nucleation complex, PtdIns3P-binding complex, conjugation system, and degradation [7] (Arrow shape indicates next step).

**Figure 2 ijms-21-02008-f002:**
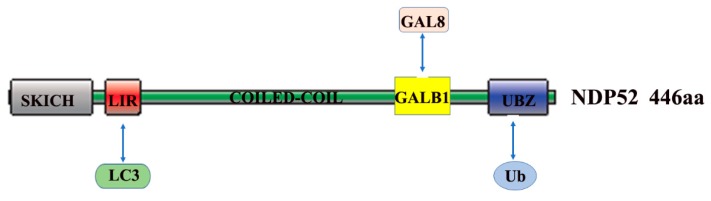
Structure-function of NDP52. NDP52 contains 446 amino acid (aa), including the domains of SKICH, LC3-interaction region (LIR), COILED-COIL, GALB1, and UBZ. UBZ domain interacts with Ub, identifies and binds autophagy substrates. LIR domain interacts with LC3 and assists substrate anchoring to autophagosome membranes. GALB1 domain interacts with GAL8 (modified from reference [30]).

**Figure 3 ijms-21-02008-f003:**
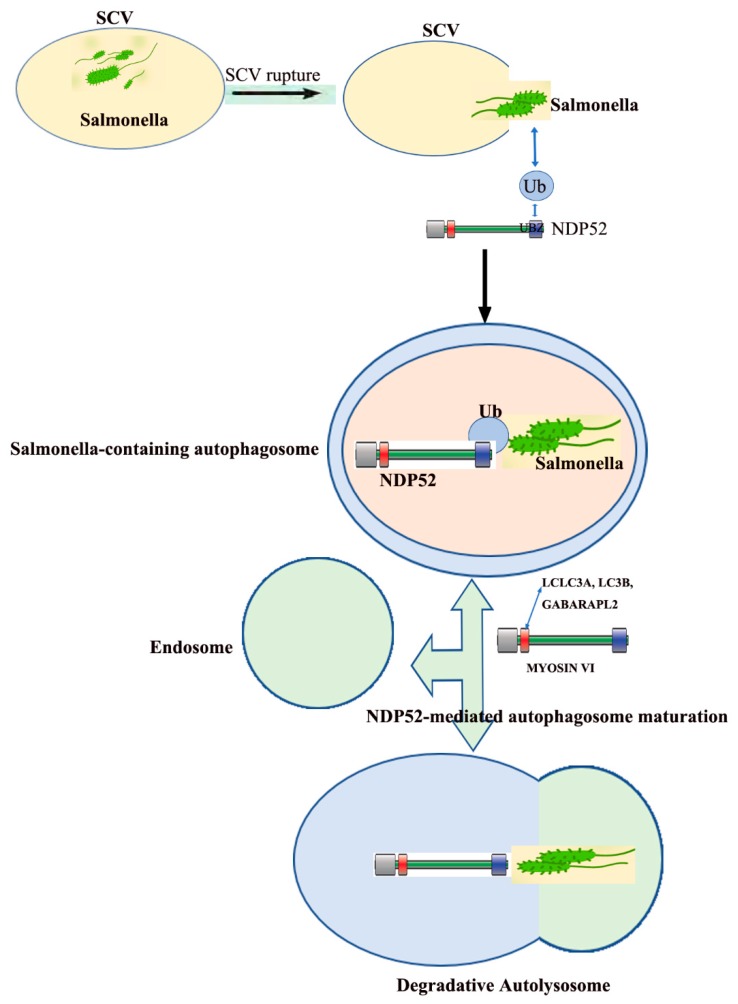
Dual role of NDP52 in *Salmonella* infection. *Salmonella* colonizes within *Salmonella*-containing vesicles (SCV) and replicates itself. After the vesicles burst, they are released into the cytoplasm and reproduce in large numbers. NDP52 identify and target *Salmonella* that are released into the cytoplasm through its UBZ domain and LC3c binding site (CLIR motif), and, independently, regulates autophagosome maturation through its LC3a, LC3b, or Gabarapl2 binding site (LIR-like motif) and its Myo6 interacting region (modified from [37]).

**Figure 4 ijms-21-02008-f004:**
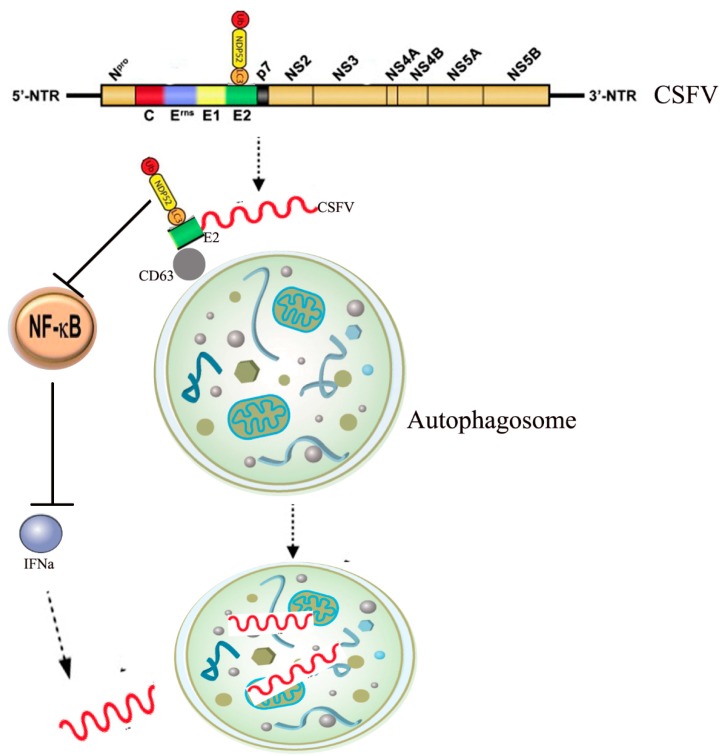
The mechanism of NDP52 in CSFV infections. In the CSFV infection PK-15 cells, NDP52 protein co-localized with CSFV E2 protein in the cytoplasm. Inhibition of NDP52 reduces the binding of E2 protein and lysosomal marker molecule CD63. The inhibition of NDP52 promotes the activation of the NF-κB signaling pathway and promotes the release of cytokines such as IFNα, inhibiting CSFV replication (Arrow shape indicates next step, T shape indicates suppression).

**Table 1 ijms-21-02008-t001:** Function of autophagy and NDP52 in bacterial infections.

Function of Autophagy and NDP52 in Bacterial Infections				
Bacteria	Induction/Inhibition Autophagy	Autophagy in Bacterial Infections	NDP52 in Bacterial Infections	Reference
*Salmonella*	Induction	Inhibition	NDP52 binds *Salmonella.*	[37]
*Shigella*	Induction	Inhibition	NDP52 targets *Shigella.*	[42,43]
*Listeria*	Induction	Inhibition	NDP52 targets the *Listeria* ActA mutant.	[43]
*Streptococcus*	Inhibition (M protein)	Inhibition	NDP52 binds the *Salmonella* ubiquitin coat.	[44]

**Table 2 ijms-21-02008-t002:** Function of autophagy and NDP52 in virus infections.

Function of Autophagy and NDP52 in Virus Infections				
Virus	Induction/Inhibition Autophagy	Autophagy in Virus Infections	NDP52 in Virus Infections	Reference
CSFV	Induction	Promotes CSFV replication	NDP52 interacts with E2 protein	[100]
MeV	Induction	Promotes Mev replication	NDP52 interacts with MeV-C and MeV-V protein	[89]
CVB3	Induction	Promotes CVB3 replication	NDP52 interacts withVP1, promoting the ubiquitination of VP1 protein	[93]
CHIV	Induction	Promotes CHIV replication	NDP52 interacts withVP1, reducing CHIV infection	[94]
Influenza Virus	Induction	Promotes Influenza Virus replication	PB1-F2 interacts with NDP52	[86]
HSV1	Inhibition	Restricts HSV1 replication	HSV1 removes NDP52 from the nucleus	[84]

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
