# Peer review of "The Role of Autophagy and Autophagy Receptor NDP52 in Microbial Infections"

_ijms, 2020, doi:10.3390/ijms21062008_

Round 1
Reviewer 1 Report
Dear review article is well written.
It needs some improvements such as:
1- The authors should explain more the meaning of liposomes in relationship to Autophagy.
2- The relationship between Autophagy receptor NDP52 and LAP in bacterial and viral infection should be explained in detail
3-Is there any relationship between this Autophagy receptor and lipophagy or mitophagy in bacterial and viral infections?
4-As Autophagy promotes viral infection, is there any treatment which could block the action of NDP52 and subsequently prevent the increase of infection?
Author Response
Point 1:The authors should explain more the meaning of liposomes in relationship to Autophagy.
Response 1:Thank you very much for your comments. After careful consideration, we think that the description "a liposome-like" structure may not be appropriate. We have replaced by “a double-membraned structure” in line41.
Point 2: The relationship between Autophagy receptor NDP52 and LAP in bacterial and viral infection should be explained in detail.
Response 2:Thank you very much for your comments, and we added some details in line of 205-209. NADPH oxidase activity play an important role in LAP degrades microorganisms and activation the ATG conjugation systems, which is dispensable for xenophagy [56]. Interestingly, during Listeria and Salmonella infection, both LAP and xenophagy occur at the same time [57], and they can be difficult to distinguish since both are defined by membrane-associated LC3 [58]. We also added some details in line of 298-302. Moreover, Streptococcus pneumoniae can trigger the formation of pneumococcus-containing LC3-associated phagosome (LAPosome)-like vacuoles (PcLVs) in an early stage of infection, which are indispensable for subsequent formation of bactericidal pneumococcus-containing autophagic vacuoles (PcAVs). The Atg16L1 WD domain, p62, NDP52, and poly-Ub contributed to PcLV formation[88].
Point 3:Is there any relationship between this Autophagy receptor and lipophagy or mitophagy in bacterial and viral infections?
Response 3:Thank you very much for your comments. Lipophagy, as any form of selective autophagy, primarily begins with the recognition of the cargo by the autophagosomal membrane through interaction with LC3. This typically involves the assistance of one or more cargo adaptors such as p62, Optineurin, NBR1 and NDP52 that connect the organelle membrane with LC3, and may require polyubiquitination of proteins on the organelle surface as a recruiting signal [1]. However, we do not know if there is a relationship between NDP52 and lipophagy in bacterial and viral infections because we have not found relevant literature.
Our study found that inhibition of NDP52 decreases mitophagy during CSFV infection, which means that NDP52 promotes mitophagy in CSFV infection [2].
Rogov, V.; Dötsch, V.; Johansen, T.; Kirkin, V., Interactions between Autophagy Receptors and Ubiquitin-like Proteins Form the Molecular Basis for Selective Autophagy. Molecular Cell 2014, 53, (2), 167-178. Fan, S.; Wu, K.; Luo, C.; Li, X.; Zhao, M.; Song, D.; Ma, S.; Zhu, E.; Chen, Y.; Ding, H.; Yi, L.; Li, J.; Zhao, M.; Chen, J., Dual NDP52 Function in Persistent CSFV Infection. Frontiers in Microbiology 2020, 10.Point 4:As Autophagy promotes viral infection, is there any treatment which could block the action of NDP52 and subsequently prevent the increase of infection?
Response 4:Thank you very much for your comments. We have found that NDP52-specific interference RNA can inhibit CSFV replication and reduce the occurrence of autophagy. In addition, we found that the ubiquitination of NDP52 plays an important role in the release of NF-kB signaling pathways and cytokines, as well as in the replication of CSFV, so we suspect that using ubiquitin inhibitors (MG-132) or the NDP52 plasmid lacking the ubiquitin domain may help prevent viral infection [2].
2.Fan, S.; Wu, K.; Luo, C.; Li, X.; Zhao, M.; Song, D.; Ma, S.; Zhu, E.; Chen, Y.; Ding, H.; Yi, L.; Li, J.; Zhao, M.; Chen, J., Dual NDP52 Function in Persistent CSFV Infection. Frontiers in Microbiology 2020, 10.
Reviewer 2 Report
This manuscript summarized the role of autophagy and NDP52 in pathogen-host interaction – an interesting topic in the field of autophagy research. In general, the authors provided a literature review on the interactions among autophagy, innate immunity pathways, and intracellular pathogens. The authors focused on the autophagy adaptor protein, NDP52, and further discussed the role of NDP52 during the infection of a number of pathogens that are clinically relevant. While the reviewer feels that this manuscript is detailed and provides a great deal of information, the quality of this manuscript is not satisfactory. The manuscript is not written and organized in a way that the readers can easily comprehend and grasp the key information. The authors may require assistance from scientific editors. Importantly, this manuscript is generally descriptive – it would be great to read about further insight into how autophagy pathways, NDP52, and pathogens interact and address why NDP52 may be exploited as a potential therapeutic target for infectious diseases. Moreover, there are numerous typos, grammatical issues, and citation errors throughout the manuscript. The reviewer made a general, but not a comprehensive list, to assist the authors on improving the quality of this manuscript for future submission
Major comments
In line 42, the authors mentioned that the formation of autophagosome begins with “a liposome-like” structure. The reviewer feels that this is not an appropriate description. “a double-membraned structure” may be a better choice.
In line 49, the authors mentioned that LC3-I is modified by ubiquitination. This is incorrect. The LC3-I is processed by a ubiquitin-like conjugation system. There are major differences between ubiquitination and ubiquitin-like conjugation system.
In line 76, the authors used the word “complete autophagy”. This is ambiguous and certainly not a standard term used in autophagy research. What defines “complete autophagy”?
In line 105, the authors made an incorrect statement that NLRP4 recruits phagosome containing Streptococcus. It should be that “phagosome containing Streptococcus recruits NLRP4”.
In line 125, the authors stated that the autophagy receptor proteins may bind to the autophagic substrates in a UBA-dependent manner. This is not entirely correct. UBA is not the only ubiquitin-binding domain required in substrate recognition during autophagy -- other ubiquitin-binding domains, such as UBAN (in OPTN) and UBZ (in NDP52) are involved as well.
In line 192, the authors mentioned that the host cells generate a series of “autoimmune responses” to clear the pathogens. This is inherently incorrect. It should be “innate immune responses” or “intracellular immune responses”.
In line 204-207, the text is almost identical to that in the abstract of ref. #44. The author should pay extra caution when quoting the original article. Moreover, a few key words in the original text are omitted, and the words “complementary” and “relies” were misspelled that the entire paragraph makes no sense.
In the Figure 3 legend, the authors noted that NDP52 target Salmonella through an “LC3C-binding site”. This is incorrect. NDP52 recognizes the ubiquitinated bacteria via a UBZ domain.
In line 254, the authors claimed that “NDP52-regulated Salmonella and Shigella can be used as a therapeutic mean” without providing any rationale. The authors should at least provide some explanations.
The reviewer noticed that the year of some references is incorrect. For example, the ref #36 should be a 2017 paper; ref #40 should be in 2013; and ref #72 should be in 2014. There other similar errors. The authors should be cautious when making the reference list.
There are too many writing errors such as wrong letter case, lack of space, and inappropriate punctuation.
In numerous occasions, the in-text reference included the first name of the first author as a part of the narrative.
The first name of the author should not be shown in the text. In a few occasions, the authors made a statement without providing reference(s). For example, line 141 and 306 (and many others).
The tables are hard to read. The author should shorten the table header and condense the information.
Minor comments:
In line 48, the author wrote: “the early synthesis of pro-LC3 is cleaved by Atg4”. The synthesis of pro-LC3 cannot be cleaved; pro-LC3 can be. Similar errors appeared many times in the manuscript and such faulty reference can be misleading.
In line 63, the authors mentioned that “autophagy selective removes proteins from damaged or cellular aging organelle”. The review feels necessary to point out that autophagy removes not only the protein but the entire damaged, unwanted organelle.
Instead of using “autophagy-related proteins” or “ATG proteins”, the authors used “autophagy-associated protein” (line 96) and “Atgs” (line 112). This is not a standard nomenclature.
Line 99: “MHCII” should be corrected as “MHC class II”
Line 201: the term “multi-ubiquitinated” should be “poly-ubiquitinated”; line 203: “the identification of Salmonella” should be “the recognition of the invaded Salmonella”.
Author Response
Response to Reviewer 2 Comments
Point 1: The manuscript is not written and organized in a way that the readers can easily comprehend and grasp the key information.
Response 1: Thank you very much for your comments. We deleted some content of the introduction about autophagy and innate immunity (original line73-113), so that the article is more compact and easier to understand the role of NDP52.
Point 2: Importantly, this manuscript is generally descriptive – it would be great to read about further insight into how autophagy pathways, NDP52, and pathogens interact and address why NDP52 may be exploited as a potential therapeutic target for infectious diseases.
Response 2: Thank you very much for your comments. We added some description on why NDP52 may be exploited as a potential therapeutic target for infectious diseases (line 241-249).
During Salmonella infection, anti-bacteria autophagy requires the participation of TBK1 and W1P12[69], and CALCOCO2 NDP52 initiates selective autophagy through recruitment of ULK and TBK1 kinase complexes[70]. In addition, NDP52 can promote maturation of Salmonella-encapsulated vesicles to promote Salmonella clearance[37]. Shigella recruits Toca-1 to inhibit the aggregation of NDP52 and LC3 to reduce xenophagy[71]. In addition, Shigella alters the NF-kB signaling pathway and inflammatory response to act on its own replication[72]. Studies have found that NDP52 promotes NF-kB activation[52], suggesting that NDP52-regulated Salmonella and Shigella may be used as therapeutic means.
Point 3: Moreover, there are numerous typos, grammatical issues, and citation errors throughout the manuscript.
Response 3: Thank you very much for your comments. We have commissioned David Sanders(d.sander@ucl.ac.uk) to help us improve our language. We re-examined and revised our references.
Major comments
Point 4: In line 42, the authors mentioned that the formation of autophagosome begins with “a liposome-like” structure. The reviewer feels that this is not an appropriate description. “a double-membraned structure” may be a better choice.
Response 4: Thank you very much for your comments. We have changed the description in line 41.
Point 5: In line 49, the authors mentioned that LC3-I is modified by ubiquitination. This is incorrect. The LC3-I is processed by a ubiquitin-like conjugation system. There are major differences between ubiquitination and ubiquitin-like conjugation system.
Response 5: Thank you for pointing out the mistakes. We have changed the description in line 50.
Point 6: In line 76, the authors used the word “complete autophagy”. This is ambiguous and certainly not a standard term used in autophagy research. What defines “complete autophagy”?
Response 6: Thank you for pointing out the mistakes. Yes, “complete autophagy” is not a standard term used in autophagy research, and we have corrected it in line 76.
Point 7: In line 105, the authors made an incorrect statement that NLRP4 recruits phagosome containing Streptococcus. It should be that “phagosome containing Streptococcus recruits NLRP4”.
Response 7: Thank you for pointing out the mistakes. We deleted some content of the introduction about autophagy and innate immunity (original line73-113).
Point 8: In line 125, the authors stated that the autophagy receptor proteins may bind to the autophagic substrates in a UBA-dependent manner. This is not entirely correct. UBA is not the only ubiquitin-binding domain required in substrate recognition during autophagy -- other ubiquitin-binding domains, such as UBAN (in OPTN) and UBZ (in NDP52) are involved as well.
Response 8: Thank you for pointing out the mistakes. We have changed UBA into ubiquitin-binding domains in line 100.
Point 9: In line 192, the authors mentioned that the host cells generate a series of “autoimmune responses” to clear the pathogens. This is inherently incorrect. It should be “innate immune responses” or “intracellular immune responses”.
Response 9: Thank you for pointing out the mistakes. We have changed autoimmune responses into innate immune responses in line 174.
Point 10: In line 204-207, the text is almost identical to that in the abstract of ref. #44. The author should pay extra caution when quoting the original article. Moreover, a few key words in the original text are omitted, and the words “complementary” and “relies” were misspelled that the entire paragraph makes no sense.
Response 10: Thank you for pointing out the mistakes. We have changed it into NDP52 interacts with LC3A, LC3B, and/or GABARAPL2 through a distinct LC3-interacting region, interacting with MYOSIN VI to promote the maturation of autophagosomes. NDP52 for autophagosome maturation is complementary but does not depend on its function in pathogen-targeted autophagosomes, which depends on interaction with LC3C(line 186-190).
Point 11: In the Figure 3 legend, the authors noted that NDP52 target Salmonella through an “LC3C-binding site”. This is incorrect. NDP52 recognizes the ubiquitinated bacteria via a UBZ domain.
Response 11: Thank you for pointing out the mistakes. We have changed NDP52 identify and target Salmonella that released into the cytoplasm through its LC3C binding site (CLIR motif) into NDP52 identify and target Salmonella that released into the cytoplasm through its UBZ domain and LC3C binding site (CLIR motif).
Point 12: In line 254, the authors claimed that “NDP52-regulated Salmonella and Shigella can be used as a therapeutic mean” without providing any rationale. The authors should at least provide some explanations.
Response 12: Thank you very much for your comments. We added some description on why NDP52 may be exploited as a potential therapeutic target for infectious diseases (line 241-249). During Salmonella infection, anti-bacteria autophagy requires the participation of TBK1 and W1P12[69], and CALCOCO2 NDP52 initiates selective autophagy through recruitment of ULK and TBK1 kinase complexes [70]. In addition, NDP52 can promote maturation of Salmonella-encapsulated vesicles to promote Salmonella clearance [37]. Shigella recruits Toca-1 to inhibit the aggregation of NDP52 and LC3 to reduce xenophagy [71]. In addition, Shigella alters the NF-kB signaling pathway and inflammatory response to act on its own replication [72]. Studies have found that NDP52 promotes NF-kB activation [52], suggesting that NDP52-regulated Salmonella and Shigella may be used as therapeutic means.
Point 13: The reviewer noticed that the year of some references is incorrect. For example, the ref #36 should be a 2017 paper; ref #40 should be in 2013; and ref #72 should be in 2014. There other similar errors. The authors should be cautious when making the reference list.
Response 13: Thank you for pointing out the mistakes. We re-examined and revised our references, and marked in red in the references.
Point 14: There are too many writing errors such as wrong letter case, lack of space, and inappropriate punctuation.
Response 14: Thank you for pointing out the mistakes. We have commissioned David Sanders(d.sander@ucl.ac.uk) to help us improve our language.
Point 15: In numerous occasions, the in-text reference included the first name of the first author as a part of the narrative.
The first name of the author should not be shown in the text. In a few occasions, the authors made a statement without providing reference(s). For example, line 141 and 306 (and many others).
Response 15: Thank you for pointing out the mistakes. We have corrected this error and added references in the appropriate places.
Point 16: The tables are hard to read. The author should shorten the table header and condense the information.
Response 16: Thank you very much for your comments. We have improved Table1 and Table2 to make it easier to read.
Minor comments:
Point 17: In line 48, the author wrote: “the early synthesis of pro-LC3 is cleaved by Atg4”. The synthesis of pro-LC3 cannot be cleaved; pro-LC3 can be. Similar errors appeared many times in the manuscript and such faulty reference can be misleading.
Response 17: Thank you for pointing out the mistakes. We have corrected this error in line 48-50.
Point 18: In line 63, the authors mentioned that “autophagy selective removes proteins from damaged or cellular aging organelle”. The review feels necessary to point out that autophagy removes not only the protein but the entire damaged, unwanted organelle.
Response 18: Thank you very much for your comments. We have changed “autophagy selective removes proteins from damaged or cellular aging organelle” into Autophagy selectively removes damaged or cellular aging organelles.
Point 19: Instead of using “autophagy-related proteins” or “ATG proteins”, the authors used “autophagy-associated protein” (line 96) and “Atgs” (line 112). This is not a standard nomenclature.
Response 19: Thank you very much for your comments. We deleted some content of the introduction about autophagy and innate immunity (original line73-113), and we have changed Atgs into ATG proteins in line 87.
Point 20: Line 99: “MHCII” should be corrected as “MHC class II”
Response 20: Thank you very much for your comments. We deleted some content of the introduction about autophagy and innate immunity (original line73-113).
Point 21: Line 201: the term “multi-ubiquitinated” should be “poly-ubiquitinated”; line 203: “the identification of Salmonella” should be “the recognition of the invaded Salmonella”.
Response 21: Thank you very much for your comments. We have changed “multi-ubiquitinated” into poly-ubiquitinated (line 183), and changed “the identification of Salmonella” into “the recognition of the invaded Salmonella” (line 185).
Reviewer 3 Report
In this review, the importance of the autophagy receptor NDP52 is detailed. I found the review to be quite comprehensive, as it discusses the role of the receptor in the autophagic response to a number of bacteria, including Salmonella, Shigella, Listeria and Streptococcus, as well as numerous viruses such as influenza, classical swine flu, measles, Coxsackie B3, Chikungunya, and Herpes simplex viruses. In the process, the review makes it clear that the receptor is multi-functional and acts in a number of different ways, depending on the particular microbe, from binding viral proteins, e.g. influenza PB1-F2, to mediating autophagic degradation of substrates. I especially like how it is made clear exactly how different microbes have exploited the receptor to their advantage.
However, while comprehensive, my primary criticism of this review is that it took a long time to get to the real meat of the topic of the role of NDP52 in the autophagic response to microbial infections. In my opinion, the introduction can be shortened considerably without compromising the reader’s understanding of the wide-ranging role of the receptor in the autophagic response to these infections.
Finally, the review needs considerable editorial attention for use of the English language.
Author Response
Point 1:However, while comprehensive, my primary criticism of this review is that it took a long time to get to the real meat of the topic of the role of NDP52 in the autophagic response to microbial infections. In my opinion, the introduction can be shortened considerably without compromising the reader’s understanding of the wide-ranging role of the receptor in the autophagic response to these infections.
Response 1: Thank you very much for your comments. We deleted some content of the introduction about autophagy and innate immunity (line73-113), so that the article is more compact and easier to understand the role of NDP52.
Point 2:Finally, the review needs considerable editorial attention for use of the English language.
Response 2:Thank you very much for your comments. We have commissioned David Sanders(d.sander@ucl.ac.uk) to help us improve our language.
Round 2
Reviewer 2 Report
The reviewer feels that this manuscript has slightly improved but has not reached the quality for publication. The overall organization has become more cohesive; however, the major issues are still there. In the revised manuscript, the author only made corresponding changes based on the reviewer’s comments but did not carefully check the entire manuscript for many other obvious issues/mistakes. The authors will require further, extensive scientific editing to improve the article.
Since the authors were not able to identify and correct most of the issues/mistakes in the revised text, the reviewer has made additional comments below and provided a PDF file with highlights. Please check the manuscript thoroughly and note that these comments are not comprehensive; it is the responsibility of the authors to identify and correct many other problems.
Comments:
Line 51-55: The description is lengthy and unclear
Line 92: “Surface structure” is not clear. Does it mean protein or specific lipid membrane microdomain?
Line 106-109: Two sentence should combine to avoid confusion, such as “The ATG5-ATG12 complex can directly interact with the CARD domain of RIG-I and IPS1 to suppress the next RLRs signal [26], suggesting that that ATG proteins may act as negative regulators of RLR-mediated antiviral response”
Line 109: The reviewer recommends moving the sentence ”NDP52, also known as….” to the beginning of the next paragraph. This sentence is much related to the next section in which NDP52 and ubiquitin-UPS system are discussed.
Line 117: The UPS and selective autophagy are involved in the clearance of intracellular pathogens, but proteasome-dependent degradation is not generally considered as a part of selective autophagy. The authors should consider rephrasing or removing this sentence.
Line 119: Ubiquitin does not cause polyubiquitination of the autophagic substrate.
Line 129: The author mentioned T6BP without providing any information or citation. Is the binding of T6BP binding to myosin VI relevant to cytokine signaling and cell adhesion?
Line 133-136: Consider merging this paragraph with the previous one.
Line 134: The sentence does not make sense. “Autophagic degradation” and “autophagic clearance of bacteria” are basically the same thing. Moreover, the reviewer is not sure about the meaning of “NDP52 is involved in the immunization and anti-epidemic process”. Immunization and anti-epidemic process may not be appropriate terms. Do the authors mean “innate immunity” or “host defense mechanisms”?
Line 287: The meaning of “Furthermore, autophagy receptors are usually targeted to most genus-specific proteins rather than the conservative proteins” is not clear. Are the autophagy receptor targeted by genus-specific proteins or the autophagy receptors target to genus-specific proteins? What is the significance of this information?
Line 306-310: These sentences do not make sense.
Line 320: “relies on autophagy binding bacteria and phagophore membrane” does not make sense.
Line 363: The sentence “Bacteria membrane residue…..” does not make sense and the relevant information is not found in the ref. 59.
Line 376: The description of IscB is unclear. The author should make clear as to whether IcsB suppresses NDP52 directly or via the inhibition of Toca-1/LC3 interaction.
Line 380: Does “a novel structural model” means “a unique structural fold”?
Line 401: “to act on its own replication” is ambiguous. Please clarify.
Line 402: The authors stated that “NDP52-regulated Salmonella and Shigella may be used as therapeutic means”. Does this statement mean that the bacteria itself may be used as a therapeutic approach? Please clarify.
Line 438: “In the absence of NDP52 and TBK1………cells” does not make sense. Please rephrase it.
Line 454: “SpeB degraded the expression of autophagy receptors” does not make sense and need to be rephrased.
Line 456-460: This piece of information is unclear.
Line 462: “NDP52 is reset in virus infection” is unclear. What does “reset” mean in this context?
Line 463-465: The entire sentence does not make sense.
Line 465-468: “Unlike HSV-1…………” this paragraph is adapted from the original article (ref.89) but the rephrased text does not make sense. Also, the reviewer thinks that the rephrased sentences share too many similarities to the original text.
Line 469: What does “improved localization of NDP52” mean? This statement is unclear.
Line 482-522: This paragraph provided a lot of information but not easy to read. Please consult with scientific editing.
Line 497-499: Please rephrase “NDP52 and p62 play…….” This sentence does not make sense.
Line 500: the statement of “ubiquitin production of the VP1 protein” is unclear.
Line 504: What is “CHIV RCS”? Do the authors mean “CHIKV replication complexes (RCs)”? If so, how could the replication complexes function to reduce the replication of CHIV itself?
Line 523: This statement is unclear and confusing. “NDP52 inhibition also inhibits CSFV replication and the induction of mitophagy”. Does that mean NDP52 inhibits mitophagy? Based on the literature, CSFV infection induces mitophagy.
Suggested edits:
The authors should check all gene nomenclature and choose consistent names (such as Atg5 or ATG5; NDP52 or CALCOCO2 NDP52)
Line 57: “substrates” --> autophagic substrate
Line 63: “Atg12 conjugation system and Atg8/LC3 conjugation system” --> “conjugation system”
Line 89: The invaded cells --> the host cell
Line 95: Remove “such as”
Line 118: “P62” --> “p62/SQSTM1”
Line 118: Here, does “intermediate” means “mediate” or “middle-level”?
Line 319: “invasion by the bacteria” --> “invading bacteria”
Line 131: “autophagy maturation” --> “autophagosome maturation”
Line 324: --> “eat-me signal initiates autophagy”.
Line 326: “poly-ubiquitinated protein” --> “poly-ubiquitination”
Line 348: --> “that are released”
Line 368: The reviewer is confused by “Atc5”. Do the authors mean “Atg5”?
Line 371: Remove “via the molecular barrier” and correct “IscsB”.
Line 374: --> “which is secreted”.
Line 375: --> “the type III system”
Line 376: “Autophagy markers” --> “autophagy machinery”
Line 396: “W1P12” should be “WIPI1/2”
Line 421: --> “by regulating PrfA and phospholipases”
Line 424: “Researchers” --> “Previous studies”
Line 432: --> “invasive strain which can cause significant mortality”
Line 434: “killed” --> “eliminated”
Line 441: --> “ATG/LC3-positive”
Line 444: --> “the expression of autophagy receptors is inhibited”
Line 480: Same as the previous comments on “complete autophagy”. Please check the entire manuscript for similar errors.
Line 522: The meaning of “PINK-Parkin 2” is unclear. Do the authors mean “PINK1/Parkin pathway”?
Author Response
Please see the attachment.

This manuscript is a resubmission of an earlier submission. The following is a list of the peer review reports and author responses from that submission.
Round 1
Reviewer 1 Report
In this review, the authors investigated the role of autophagy receptor NDP52 in microbial infections. There major defects in this manuscript:
1-The figures are not clear and needs massive improvements, specifically figures 2 and 4. Where is the figure legend of Fig. 4??
Also, there is disconnections between figures and text...you should refere to the figures in text.
2-The authors supported their story related to the autophagy in virus infections by unpublished data....you should show support from published data.
3-It is better to show a table the various bacterial and viral infections associated with the role of autophagy and NDP52 in every infection.
Reviewer 2 Report
This review article summarized mainly the roles and interactions of NDP52, an autophagy receptor, within the infection of various pathogens, including bacteria and viruses. The content of paper contributes to our understanding of the roles of NDP52 in microbial infection. However, There are some major points listed below that need to be addressed or improved:
Review points:
All figures are not mentioned in the text. The “introduction” and “NDP52” sections need to be reorganized. Name of bacteria should be italic. Figures 1 to 3 are adapted from Figure 2, Figure 2A, and graphical abstract of the original papers, respectively, without permission indicated. The legend of Figure 2 is almost identical to the one in the original figure. In the legend of Figure 3 (page 5, line 26), CALCOCO2 should be replaced by NDP52 because there is no CALCOCO2 in the figure. There is no detailed description in the legend of Figure 4. The content of the last paragraph on Page 8 (lines 33-48) is not complete, and its format is not consistent in the text. The manuscript needs to be proofread by a native English speaker to eliminate errors in English.